



# Process Rate Estimator: A novel model to predict total denitrification using natural abundance stable isotopes of N₂O

Charlotte Decock[1,2], Juhwan Lee[2,3], Matti Barthel[2], Elizabeth Verhoeven[2,4], Franz Conen[5], Johan Six[2]

[1]College of Agriculture, Food and Environmental Sciences, California Polytechnic State University, San Luis Obispo, CA 93407, USA
[2]Department of Environmental Systems Science, ETH Zurich, 8092 Zurich, Switzerland
[3]Department of Smart Agro-industry, Gyeongsang National University, Jinju-si, Gyeongsangnam-do, 52725, Republic of Korea
[4]Department of Crop and Soil Science, Oregon State University, 3017 Agriculture and Life Sciences Building, Corvallis, OR 97331, USA
[5]Department of Environmental Sciences, University of Basel, Bernoullistrasse 30, 4056 Basel, Switzerland

*Correspondence to:* Charlotte Decock (cdecock@calpoly.edu)

**Abstract.** Total denitrification, the natural process capable of removing reactive N from ecosystems through conversion to N₂, is one of the most poorly constrained processes in terrestrial N cycling. In situ quantification of total denitrification could help identify mitigation options for N pollution. This study provides proof-of-concept for a novel natural abundance isotope based model for depth differentiated in situ quantification of total denitrification; it does so by examining the use-case of the impact of wheat (Triticum aestivum) varieties with different root biomass on total denitrification. We set up a mesocosm experiment in 1.5m tall lysimeters with four wheat varieties, each replicated three times. Temporal data for soil moisture, nitrous oxide (N₂O) concentrations in the soil pore space, site preference (SP) and δ¹⁸O values of soil pore space N₂O were collected at soil depths of 7.5, 30, 60, 90 and 120 cm over a five month growing period and used as input variables in the new model. Here, we define total denitrification as gross N₂O consumption, with N₂O produced either through nitrification or denitrification. The model, further referred to as 'Process Rate Estimator' or PRE, constrains temporally explicit gross N₂O production and consumption rates at each depth increment based on a combination of diffusion and isotope mixing and fractionation models. Estimated production and consumption of N₂O, integrated over the five month experiment, ranged from 3.9 to 170.3 kg N ha-1, with a trend for greater N₂O production from denitrification compared to nitrification. N₂O concentrations where greatest at 60 and 90 cm depth, while N₂O production and consumption peaked at 7.5 and 30 cm depth, illustrating the important role of N₂O dynamics along the soil profile in understanding ecosystem N budgets. Both N₂O production and consumption were greater in varieties that had previously been characterized to have greater root biomass. We demonstrate that PRE is able to constrain nitrification and denitrification leading to gross daily N₂O production, and gross reduction to N₂ across the depth profile, based on the temporal change in concentrations, δ¹⁸O and SP of N₂O. We conclude that our results support the potential of PRE to estimate total denitrification in situ, which could form the basis for developing promising strategies to reduce N pollution.





## 1 Introduction

Since the start of the green revolution, humans have relied on the Haber-Bosch process to fix atmospheric dinitrogen gas (N2) into plant available forms, boosting animal feed and human food production. With current N fixation rates estimated at 121 million tons per year, however, the amount of reactive N present in our ecosystems exceeds 3.5 times the boundary considered as a safe operating space for humanity (Rockström et al., 2009). Denitrification, defined as the reduction of nitrate ($NO_3^-$) to $N_2$ by microorganisms, is arguably our most powerful tool to remove reactive N from ecosystems, thereby reducing N emissions and leaching. Denitrification is ubiquitous across ecosystems, exhibiting a high level of both functional diversity and functional redundancy across a range of phylogenetically diverse organisms, with a wide range of environmental tolerances (Zumft, 1997). In fact, denitrification is known to successfully fuel the removal of reactive N in wastewater treatment plants, bioreactors, constructed wetlands, and riparian areas (Lu et al., 2014; Schipper et al., 2010; Hang et al., 2016; Martin et al., 1999). Given these success stories, denitrification may be an effective strategy to remove excess N in agricultural ecosystems. Moreover, a better understanding of denitrification rates in agroecosystems would help close ecosystem N budgets and fine-tune N management plans. Despite glaring opportunities, rates of denitrification from the soil profile are poorly constrained, mainly due to methodological limitations (Groffman et al., 2006; Schlesinger, 2009).

On a global scale, total $N_2$ emissions from denitrification were estimated to range between 67 to 130 million tons N per year, with roughly half of this N removal attributed to agricultural soil (Scheer et al., 2020).  Daily gross denitrification rates from soil ranged between 0.001 and 20 kg N ha$^{-1}$ day$^{-1}$ across studies, correlating positively to soil water-filled pore space, nitrate ($NO_3^-$) content, and temperature, and negatively with soil oxygen content (Pan et al., 2022).  When studying denitrification, potential emissions of the potent greenhouse gas and ozone depleting substance nitrous oxide ($N_2O$) need to be considered (Ipcc, 2013; Ravishankara et al., 2009), as $N_2O$ is an intermediate compound formed during the reduction of $NO_3^-$ to $N_2$ by denitrifiers (Zumft, 1997). In addition, $N_2O$ can be formed as a byproduct of nitrification, or through a process referred to as nitrifier denitrification, which involves the reduction of $NO_2^-$ by ammonia oxidizing bacteria (Wrage et al., 2004). Regardless of the source process, $N_2O$ can be further reduced to $N_2$ by denitrifiers (Jones et al., 2013). In this context, a commonly used metric is the ratio of $N_2O:(N_2O+N_2)$, where a lower ratio indicates a greater importance of gross $N_2O$ consumption or total denitrification in an ecosystem (Schlesinger, 2009; Stevens et al., 1998). Ratios of $N_2O/(N_2O+N_2)$ were found to increase with soil oxygen content but decrease with soil organic carbon (C), C:N ratio, soil pH and soil water filled pore space (Pan et al., 2022). Furthermore, total denitrification was shown to not only be affected by the quantity of organic C added or present in the soil, but also its biochemical composition (Henry et al., 2008; Morley and Baggs, 2010). While various studies have shown relative changes in $N_2O:(N_2O+N_2)$ ratios by agricultural management practices using laboratory proxies, the quantification of total denitrification under field conditions could greatly facilitate a more intentional approach to managing denitrification in crop production.

Over the past decades, various appraoches have been used to estimate total denitrification rates. These include, but are not limited to, acetylene-based inhibition methods, $^{15}N$ tracers, direct $N_2$ quantification, $N_2$:Ar ratio quantification, mass balance





approaches, methods based on natural abundance stable isotopes of nitrous oxide (N₂O), and molecular approaches (Groffman et al., 2006; Friedl et al., 2020). To date, none of these methods have adequately quantified total denitrification (gross N₂O consumption) under field conditions over extended periods of time. Acetylene inhibition, N₂:Ar ratio quantification and molecular approaches merely provide insights into potential denitrification and are not effective at quantifying absolute denitrification rates (Friedl et al., 2020; Gallarotti et al., 2021). ¹⁵N tracers have been applied under field conditions, but were shown to underestimate total denitrification due to difficulties of homogenous mixing of tracers with substrate pools for denitrification in soil microsites (Wen et al., 2016). Moreover, ¹⁵N tracers to quantify N₂ fluxes can only be used for a snapshot in time, as ¹⁵N tracers mixed with substrate pools turn over rather rapidly, at which point quantification of ¹⁵N₂ fluxes is jeopardized. Direct measurements of N₂ emitted from soil in a N₂-free atmosphere is likely the most accurate strategy to quantify total denitrification at present (Wen et al., 2016; Wang et al., 2020). Unfortunately, this method is limited in scope and application because it can only be executed under controlled laboratory conditions. Even then it is very difficult to prevent ingress of atmospheric N₂. Morover, extracting soil samples and bringing them into the laboratory environment is associated with sample disturbance and exposure to altered environmental conditions, both of which can impact denitrification rates. In contrast, natural abundance isotopes of N₂O can be analyzed on gas samples collected *in situ* over extended periods of time, and have been suggested as a promising tool to quantify total denitrification under field conditions (Lewicka-Szczebak et al., 2017; Verhoeven et al., 2019).

The use of natural abundance isotopes to estimate total denitrification, defined here as gross N₂O consumption, relies on the imprint of N₂O production and consumption processes on the isotope values of N₂O. The most frequently studied natural abundance isotope values of N₂O include bulk $\delta^{15}N$, $\delta^{18}O$ and site preference (SP), defined as the difference in isotope value between the central and terminal N atom in the N₂O molecule (Toyoda and Yoshida, 1999; Decock and Six, 2013c). Considering nitrification, denitrification and the reduction of N₂O to N₂ as the three main proceses influencing soil emitted N₂O and N₂, various studies have characterized isotope effects associated with these processes (Perez, 2005; Sutka et al., 2006; Ostrom et al., 2007; Well and Flessa, 2008; Well and Flessa, 2009; Snider et al., 2012; Snider et al., 2013). While isotope effects of nitrification, denitrification and N₂O reduction to N₂ on $\delta^{15}N_{bulk}$ of N₂O are relatively well characterized, interpretation of $\delta^{15}N_{bulk}$ values of N₂O is contingent on isotope value of the substrates $NH_4^+$ and $NO_3^-$ (Perez, 2005), which can vary drastically over time and are cumbersome to measure (Decock and Six, 2013a). Technological advances that facilitated the measurement of SP brought new hope for source partitioning N₂O, as it was discovered that nitrification and denitrification derived N₂O have unique SP values that are independent from isotope values of the substrate (Toyoda et al., 2002; Ostrom and Ostrom, 2011). Nevertheless, site preference was also shown to be affected by N₂O consumption (Jinuntuya-Nortman et al., 2008; Well and Flessa, 2009). Therefore, under field conditions where N₂O reduction to N₂ is common, SP alone cannot resolve source processes underlying N₂O production. Relationships between $\delta^{15}N$, $\delta^{18}O$ and SP observed during the reduction of N₂O to N₂ led to the use of graphical mapping approaches to provide semi-quantitative insights into the source processes underlying N₂O emissions (Toyoda et al., 2011; Opdyke et al., 2009; Köster et al., 2011). Meanwhile, research on





the dynamics of $^{18}O$ during $N_2O$ production and consumption elucidated $\delta^{18}O$ endmembers characteristic of nitrification and denitrification derived $N_2O$, taking into account oxygen exchange with water (Snider et al., 2012; Snider et al., 2013; Lewicka-Szczebak et al., 2017; Kool et al., 2011). A dual isotope appraoch based on $\delta^{18}O$ and SP values of $N_2O$, was proposed to simultaneously estimate the the contribution of nitrification and denitrification to gross $N_2O$ production and the fraction of

$N_2O$ reduced to $N_2$ in $N_2O$ emitted from the soil surface (Lewicka-Szczebak et al., 2017). Gross $N_2O$ consumption can then be estimated based on surface $N_2O$ fluxes and the fraction of $N_2O$ reduced to $N_2$. When using the dual natural abundance isotope approach to estimate source processes of $N_2O$ in soil pore air samples collected at different soil depths in a rice field, however, the fraction of $N_2O$ reduced to $N_2$ was found to change over the depth profile (Verhoeven et al., 2019). This not only demonstrates the importance of subsurface processes and the role of microsites in understanding $N_2O$ dynamics, but also

highlights the necessity of assessing $N_2O$ diffusion fluxes and the fraction of $N_2O$ reduced to $N_2$ over the soil profile to quantify total denitrification in terrestrial ecosystems. Moreover, the fraction of $N_2O$ reduced to $N_2$ has been shown to fluctuate over time (Wang et al., 2020; Verhoeven et al., 2019), illustrating that any estimate of total denitrification at the ecosystem scale should take temporal aspects into consideration. Therefore, we postulate that a time-explicit approach using a combination of isotope and diffusion models is a promising strategy to quantify total denitrification *in situ*.

The objective of this study is to provide proof-of-concept for a novel model, further referred to as 'Process Rate Estimator' or PRE, that estimates total denitrification rates across the soil profile at discrete time points over an extended time period, by combining $N_2O$ diffusion with isotope mixing and fractionation models using a $\delta^{18}O$ and SP dual isotope approach. To this end, we set up a mesocosm experiment with four winter wheat (*Triticum aestivum*) variaties known to differ in root biomass production. Temporal data for soil moisture, nitrous oxide ($N_2O$) concentrations in the soil pore space, site preference (SP)

and $\delta^{18}O$ values of soil pore space $N_2O$ were collected at five different soil depths throughout the growing season. Here, we present details on the PRE model design and findings.

## 2. Materials and Methods

### 2.1 Mesocosm experiment

This proof-of-concept study uses data collected from a mesocosm experiment where four wheat varieties were grown in non-

weighted lysimeters under greenhouse conditions. A detailed description of the experimental design can be found in Van de Broek et al. (2020). In short, the experiment was set up as a randomized complete block design, with four treatments (four varieties of winter wheat) and three replications, using 12 non-weighted cylindrical polyethylene lysimeters with a depth of 1.5 m and a diameter of 0.5 m. The lysimeters were set up in a greenhouse with an adjustable floor that was set to 1.5 m below ground level, aligning the top of the lysimeters with the aboveground glassed-in part of the greenhouse. At five depths in each

lysimeter, namely 7.5, 30, 60, 90 and 120 cm from the soil surface, sampling ports were installed for a soil moisture probe (Decagon EC-5 capacitance-domain probe, Decagon Devices, Inc., USA) and a custom-built pore gas sampler. Soil moisture



data was recorded at 30 minute intervals using a data logger (Decagon EM-50 data logger, Decagon Devices, Inc., USA). Volumetric moisture contents were converted to water-filled pore space (WFPS) by dividing the volumetric moisture content by the total porosity. Total porosity was calculated as 1 – bulk density/2.65, with 2.65 a default factor for particle density in g

cm$^{-3}$. Average bulk density across all lysimeters and soil depths was 1.73 g cm$^{-3}$, and no significant differences between lysimeters or depths were observed. The pore gas sampler consisted of a polypropylene capillary membrane impermeable to water (Diameters Outer (OD) / Inner (ID): 7.5/5.5 mm, Length (L): 150 mm) (3M-Membrana, Germany) with one end sealed and the other connected by heat shrink plastic to a polyethylene transport line (OD/ID: 6/4 mm, L: 300 mm) terminated with a stopcock Luer-Lock connector to a 25 G needle. A 10 cm layer of gravel covered with a 2 mm water-permeable felt layer

separated the soil from the bottom of the lysimeter. Water was allowed to freely drain from the bottom of the lysimeter. Automated flux chambers were mounted on top of the lysimeter. The flux chambers were 0.5 x 0.5 x 0.15 meter with the option to add one or two extensions of 0.5 x 0.5 x 0.5 m, forming a flexible headspace height of 0.15, 0.65 or 1.15 m according to the plant height.  A stainless-steel frame supported the acrylic flanks, and the chambers opened and closed automatically with pneumatically controlled doors.

The lysimeters in the mesocosm platform were repacked with an arable soil from Estavayer-Le-Lac in Fribourg, Switzerland. Prior to filling lysimeters, soil was sifted to approximately 1 cm crumbs and homogenized. The top 15 cm were filled with soil from the original A-horizon, while the rest of the column was filled with subsoil. Both topsoil and subsoil had 21% clay, and 21 % silt.  We selected four varieties from the Swiss winter wheat (*Triticum aestivum*) breeding line; Mont Calme 268 (MC 268, introduced in Switzerland in 1926), Probus (1948), Zinal (2003) and CH Claro (2007). All four varieties were commonly

cultivated in Switzerland following their release. Previous research demonstrated differences in root biomass between the varieties (Van De Broek et al., 2020; Friedli et al., 2019).  The wheat plants were germinated from seed and vernalized before transplanting in the mesocosms at a plant density of 387 plants m$^{-2}$, which is within the range of typical agronomic practice in Switzerland (Baloch et al., 2003; Van De Broek et al., 2020). Winter wheat was grown in the greenhouse for 5 months, from 24 August 2015 to 1 February 2016. The greenhouse temperature was set to reach 20 ºC during the day and 15 ºC at night.

There was uneven maturing of plants within and between lysimeters, but plants in all lysimeters had reached flowering stage by the end of the experiment.  Fertilizer and pest management during cultivation aimed to mimic as well as possible typical agronomic management for winter wheat cultivation in Switzerland (Agridea, 2009). Coinciding with stem elongation in most of the lysimeters, we applied fertilizer at a rate of 84 kg N ha$^{-1}$, 36 kg P$_2$O$_5$ ha$^{-1}$, 48 kg K$_2$O ha$^{-1}$ and 9 kg Mg ha$^{-1}$. On 4 December 2015, coinciding with the emergence of the flag leaf in most of the lysimeters, we applied a second fertilization

with 56 kg N ha$^{-1}$, 24 kg P$_2$O$_5$ ha$^{-1}$, 32 kg K$_2$O ha$^{-1}$ and 6 kg Mg ha$^{-1}$. Fertilizer products included a compound fertilizer (Polydor, 8-13-30-1.5Mg), ammonium nitrate (27.5-0-0) and Triple super phosphate (46-0-0).





## 2.2 Concentrations and isotope values of N₂O in the soil profile

Pore air samples were collected by attaching a pre-evacuated 110 mL serum crimp vials to the pore air samplers and letting
the sample air equilibrate overnight. After removing the vials, they were over-pressurized with $N_2$ gas to prevent atmospheric air from leaking in, and a 12 mL subsample was collected for $N_2O$ analysis. Pressures before and after the addition of $N_2$ gas were recorded to correct $N_2O$ concentrations for the dilution effect. The concentration of $N_2O$ in the soil air samples were determined on a gas chromatograph equipped with a micro-electron capture detector (EDC) (Bruker 456-GC, Germany) together with a suite of standards covering the expected range in concentrations. $N_2O$ concentrations in the soil pore air were
expressed in kg $N_2O$-N per hectare in each soil layer based on the air-filled pore space (100 – WFPS (%)) observed at each sampling time.

After subsampling for $N_2O$ concentration analysis, the 110 mL serum crimp vials were analyzed for the isotopic composition of $N_2O$ using an IRMS (IsoPrime100, Elementar, Manchester, UK) coupled to a preparation unit (Trace Gas, Elementar, Manchester, UK). IRMS calibration was done using three sets of two working standards (~3 ppm $N_2O$ mixed in synthetic air)
with different isotopic compositions ($\delta^{15}N^{\alpha}$ = 0.95 ± 0.12‰ and 34.45 ± 0.18‰; $\delta^{15}N^{\beta}$ = 2.57 ± 0.09‰ and 35.98 ± 0.22‰; $\delta^{18}O$ = 39.74 ± 0.05‰ and 38.53 ± 0.11‰), which were analyzed together with each batch of 20 samples. For a more detailed instrument description please refer to Gallarotti et al. (2021).

## 2.3 N₂O surface fluxes

A dual pulsed quantum cascade laser absorption spectrometer (model CW-QC-TILDAS-SC-D), Aerodyne Research Inc.,
Billerica, MA, US) was used to measure surface $N_2O$ fluxes by connecting the instrument in a closed loop with the automated chambers on top of each lysimeter (Harris et al., 2020). The instrument was operated in flow through mode at a flow rate of 1 L/min and each chamber measured separately using a 16-port VICI valve. Before injection of gas into the measurement cell, gas was directed through a CO catalyst (Sofnocat 423, Molecular Products Limited, Harlow, UK) and a permeation Nafion dryer (PD-50T-24MSS, Perma Pure, Lakewood, NJ, USA) to scrub CO and dehumidify the sample gas. Data acquisition
(acquiring absorption spectrum) and instrument control, including VICI valve operation and scheduling was done via Aerodynes in-house software TDL Wintel. Parallel chamber closure and opening was done via custom written LabView interface (National Instruments, Austin, TX, USA) and a communication module (I-7561, ICP DAS, Hukou, Taiwan), controlling chamber opening and closure via relays (I-7067, ICP DAS, Hukou, Taiwan) activating pneumatic valves (Festo, Esslingen, Germany). All connection were done using 1/8" Teflon tubing (Swagelok, Solon, OH, USA).
At the beginning of each hour, a spectral background correction was initiated for 2 min using high purity $N_2$ (Alphagaz II, Carbagas, Rümlang, Switzerland) eliminating white noise interference from the spectral fitting. Cumulative $N_2O$ emissions over the experimental period were determined by linear interpolation of diurnal $N_2O$ fluxes measured over the course of the experiment.



### 2.4 Initial soil $NO_3^-$ content

Initial soil $NO_3^-$ content in the soil profile at the start of the experiment was determined based on $NO_3^-$ concentrations in soil pore water, soil moisture content at the time of sampling, and soil bulk density. To sample soil pore water, we applied a pressure of 8 kPa to ceramic ceramic cups (Prenart mini, Prenart Equipment APS, Denmark) installed in each lysimeter at each depth increment. Soil pore water samples were collected into 50mL centrifuge tubes equipped with gas-tight lids and tubing connectors. The vacuum intended to maximize the amount of soil pore water collected, while staying within the range of

pressures typically observed in soils. Pore water collection vessels were allowed to fill overnight, and water samples were stored in a cold room at 4°C until further analysis. Concentrations of $NH_4^+$ and $NO_3^-$ in soil pore water were determined colorimetrically using the Berthelot and Vanadium(III) chloride reduction method, respectively (Forster, 1995; Doane and Horwath, 2003). Concentrations of $NH_4^+$ were below detection limit ($< 0.1$ µg N g$^{-1}$ soil), and are therefore not reported here.

### 2.5 PRE model description

#### 2.5.1 Model concept

Conceptually, our PRE is designed to simultaneously estimates gross $N_2O$ production and consumption rates at each discrete time step in a time series by solving a set of equations describing infinitesimal changes in concentrations and natural abundance isotope values (d$^{18}$O and SP) of $N_2O$. As such, process rates estimated in each time step are semi-independent. Surface $N_2O$ fluxes are shown in this study for the purpose of data interpretation, but were not used in the model. Soil bulk density,

temporally explicit soil moisture data, $N_2O$ concentrations in the soil profile and associated d$^{18}$O and SP values of $N_2O$ are the only experimentally determined input data essential for PRE. First, diffusion fluxes between soil layers are calculated for each time point using Fick's law and subsurface pore air $N_2O$ concentration gradients between depth increments (Verhoeven et al., 2018). Next, the infinitesimal changes in concentrations and natural abundance isotope values of $N_2O$ are determined as the first derivative of a smooth curve to a time series of the observed data in each time point. Subsequently, a spectral method with

globalization using non-monotone line search is used to numerically solve a system defined by linear equations for each time step (Varadhan and Gilbert, 2009). After selecting converged solutions, PRE returns a time series of gross $N_2O$ production and consumption rates. The model was written in R and is available upon request. A detailed description of PRE follows.

### 2.5.2 Calculating diffusion fluxes

Diffusion fluxes between soil increments were calculated using Fick's law described in Eq. (1), following the procedure
outlined in Verhoeven et al. (2018):

$$F_{calc} = D_s \rho \frac{dC}{dZ}, \tag{1}$$

where $D_s$ is the gas diffusion coefficient (m$^2$ s$^{-1}$), $\rho$ is the gas density of $N_2O$ ($1.26 \times 10^6$ mg $N_2O$-N m$^{-3}$), and dC/dZ is the $N_2O$ concentration gradient from the lower depth to the upper depth (m$^3$ m$^{-4}$). Fluxes were calculated based on $N_2O$ concentration gradients between 105-135 cm, 75-105 cm, 45-75 cm, 15-45 cm, and 0-15 cm depth layers, and ambient air



above the soil surface. Atmospheric $N_2O$ was set to 0.2496 ppmv, the mean $N_2O$ concentrations measured in the greenhouse.

Fluxes were calculated in mg $N_2O$-N $m^{-2}$ $s^{-1}$ and converted to g $N_2O$-N $ha^{-1}$ $day^{-1}$.

We calculated $D_s$ following Eq. (2) based on the soil volumetric water content ($\theta_w$), air-filled porosity ($\theta_a$), the total soil porosity ($\theta_T$), the diffusivity of $N_2O$ in water ($D_{fw}$, equation 4) and air ($D_{fa}$, equations 5), and the solubility of $N_2O$, according to the Millington and Quirk relationship (Mccarthy and Johnson, 1995; Yano et al., 2014; Millington and Quirk, 1960). $\theta_T$ was

determined based on the measured bulk density, assuming a mineral particle density of 2.65 g $cm^{-3}$. We then used $\theta_T$ with direct measurements for $\theta w$ to calculated $\theta a$. The equation for $D_s$ used in this study also includes both water and air diffusion coefficients, respectively, referred as $D_{fw}$ and $D_{fa}$.

$$D_s = \left( \frac{\theta_w^{10/3} + D_{fw}}{H} + \theta_a^{10/3} \times D_{fa} \right) \times \theta_T^{-2}, \tag{2}$$

where H represents a dimensionless form of Henry's solubility constant (H′) for $N_2O$ in water at a given temperature (Pa $m^3$

$mol^{-1}$). H' for $N_2O$ is calculated using Eq. (3):

$$H' = (8.5470 \times 10^6)\exp\left(\frac{-2284}{T}\right). \tag{3}$$

The dimensionless Henry's solubility constant (H) is obtained by dividing H' by R x T where R is the gas constant (8.3145 Pa $m^3$ $mol^{-1}$ $K^{-1}$). Under standard temperature condition, a constant soil temperature of 298 K was assumed.

$D_{fw}$ for $N_2O$ was calculated using Eq. (4) (Versteeg and Van Swaaij, 1988):

$$D_{fw} = (5.07 \times 10^{-6})\exp\left(\frac{-2371}{T}\right), \tag{4}$$

where T is the soil temperature, assumed to be 298 K. $D_{fa}$ for $N_2O$ at air pressure, P (Pa) and soil temperature T (K) is calculated using Eq. 5:

$$D_{fa} = D_{fa,NTP} \times \left(\frac{T}{273.15}\right)^n \times \left(\frac{101,325}{P}\right), \tag{5}$$

where $D_{fa,NTP}$ is $0.1436 \times 10^{-4}$ $m^2$ $s^{-1}$, representing the free air diffusion coefficient under standard conditions (273.15K and

101,325Pa) and n is an empirical parameter, set to 1.81 for $N_2O$ (Massman, 1998). P represents the actual pressure, which was assumed to be 101,325Pa assuming the standard condition.

### 2.5.3 Fitting smooth curves to measured input data and determining derivatives

PRE includes a routine to fit smooth curves to measured input data in time-series, evaluate the sum of square error of fit, and compare smoothing functions. All data values are averaged at each time step of measurements and then indexed by time. The

augmented Dickey-Fuller tests was included to check if a mean time series was stationary. The smoothing functions we tested for each depth, column, and response variable include linear regression, local polynomial regression, polynomial regression, and natural or basic cubic splines. After selecting the most suitable smoothing function, time-series bootstrapping is performed on the measured time series with the iteration of 1000 to estimate uncertainty in measured input data. A new data frame is created with predicted values and first derivatives and standard deviation for each time point in the desired time series.



### 2.5.4 State function set

PRE includes three-state functions, describing the change in $N_2O$ concentrations in Eq. (6), $\delta^{18}O$ in Eq. (7) and SP in Eq. (8) over time. The change in $N_2O$ concentrations in each depth increment over time ($d(N_2O_{conc})/dt$) depends on the flux of $N_2O$ entering the depth increment from the top or the bottom through diffusion ($F_{top,in}$ and $F_{bot,in}$, respectively), the flux of $N_2O$ leaving the depth increment through diffusion ($F_{out}$), the rate of $N_2O$ produced through nitrification ($N_2O_{nit}$), the rate of $N_2O$ produced through denitrification ($N_2O_{den}$), and the rate of $N_2O$ reduced to $N_2$ ($N_2O_{red}$):

$$\frac{d(N_2O_{conc})}{dt} = F_{top,in} + F_{bot,in} - F_{out} + N_2O_{nit} + N_2O_{den} - N_2O_{red}. \tag{6}$$

The associated change in site preference ($d(SP)/dt$) can be described based on the SP of $N_2O$ in that depth increment at the start of the time step ($SP_0$), the rate and isotope values of incoming product, and the rate and isotope fractionation effect of outgoing product:

$$\frac{d(SP)}{dt} = \frac{\begin{matrix} F_{top,in}(SP_{top,in} - \eta_{SP,dif} - SP_0) + F_{bot,in}(SP_{bot,in} - \eta_{SP,dif} - SP_0) \\ + N_2O_{nit}(SP_{nit} - SP_0) + N_2O_{den}(SP_{den} - SP_0) \\ - (\eta_{SP,dif}F_{out} + \eta_{SP,red}N_2O_{red}) \end{matrix}}{N_2O_{conc,0}}, \tag{7}$$

where $SP_{top,in}$ and $SP_{bot,in}$ are the site preference values of $N_2O$ measured at that time step in the depth increment above and below, respectively; and $\eta_{SP,dif}$ is the SP isotope effect for diffusion (Well and Flessa, 2008). Thus, $(SP_{top,in} - \eta_{SP,dif})$ and $(SP_{bot,in} - \eta_{SP,dif})$ represent the SP value of $N_2O$ entering the depth increment via diffusion. $SP_{nit}$ and $SP_{den}$ represent the site preference values associated with nitrification and denitrification derived gross $N_2O$ production, while $\eta_{SP,red}$ is the SP isotope effect associated with $N_2O$ reduction to $N_2$ (Verhoeven et al., 2019; Lewicka-Szczebak et al., 2017). $N_2O_{conc,0}$ is the $N_2O$ concentration in that depth increment at the beginning of the time step.

Similar to SP, the change in $d^{18}O$ over time ($d(d^{18}O)/dt$) in each time point and depth increment can be described as:

$$\frac{d(\delta^{18}O)}{dt} = \frac{\begin{matrix} F_{top,in}(\delta^{18}O_{top,in} - \eta_{18O,dif} - \delta^{18}O_0) + F_{bot,in}(\delta^{18}O_{bot,in} - \eta_{18O,dif} - \delta^{18}O_0) \\ + N_2O_{nit}(\delta^{18}O_{nit} - \delta^{18}O_0) + N_2O_{den}(\delta^{18}O_{den} - \delta^{18}O_0) \\ - (\eta_{18O,dif}F_{out} + \eta_{18O,red}N_2O_{red}) \end{matrix}}{N_2O_{conc,0}}, \tag{8}$$

where $d^{18}O_0$ is the $d^{18}O$ value of $N_2O$ in that depth increment at the start of the time step; $d^{18}O_{top,in}$ and $d^{18}O_{bot,in}$ are the $d^{18}O$ values of $N_2O$ measured at that time step in the depth increment above and below, respectively; $\eta_{18O,dif}$ is the $^{18}O$ isotope effect for diffusion (Well and Flessa, 2008); $d^{18}O_{nit}$ and $d^{18}O_{den}$ represent the $d^{18}O$ values associated with nitrification and denitrification derived gross $N_2O$ production, while $\eta_{18O,red}$ is the isotope effect for $^{18}O$ associated with $N_2O$ reduction to $N_2$ (Verhoeven et al., 2019; Lewicka-Szczebak et al., 2017). A detailed deduction of Eq. (7) and Eq. (8) can be found in Appendix A.





Reference values for isotope effects and end members are summarized in Table 1. Note that endmembers for $N_2O_{den}$ include
heterotrophic bacterial denitrification as well as nitrifier denitrification, while end members for $N_2O_{nit}$ include bacterial
nitrification and fungal denitrification (Denk et al., 2017; Verhoeven et al., 2019; Lewicka-Szczebak et al., 2017; Decock and
Six, 2013c). Processes other than those considered in this study may contribute to gross $N_2O$ production and consumption
(Butterbach-Bahl et al., 2013), however, nitrification, denitrification and $N_2O$ reduction to $N_2$ by bacterial denitrifiers are
expected to be dominant in agricultural soils. Moreover, isotope effects or endmembers are not available for all alternative
source processes. We argue that for *in situ* quantification of total denitrification using stable isotopes of $N_2O$, considering
nitrification, denitrification and $N_2O$ reduction by bacterial denitrifiers is reasonable.

### 2.5.5 Solving the state functions of a soil system

For each time step, PRE solves the state functions, using the infinitesimal change in $N_2O$ concentrations, SP, $d^{18}O$, and
diffusion fluxes in that time point as main model constraints (model inputs), while $N_2O_{nit}$, $N_2O_{den}$ and $N_2O_{red}$ are the main
model outputs to be estimated. The model was written in R (R Core Team, 2022). Specifically, the set of state functions was
solved using the MultiStart function of the package BB (Varadhan and Gilbert, 2009). The numerical solver is based on the
Barzilai-Borwein gradient method developed by Raydan (1997), which we adopted for solving large systems of equations with
low memory use. Given that numerical solvers tend to be affected by starting values, we selected a function that iteratively
searches for solutions with different sets of starting values. The solver relies on a matrix of starting values defined by the user.
We set starting values for $N_2O_{nit}$, $N_2O_{den}$ and $N_2O_{red}$ to range between 0 and 40 g N ha$^{-1}$ day$^{-1}$ as the starting moment. In some
cases, starting values ranges had to be expanded to reach 200 g N ha$^{-1}$ day$^{-1}$ for the convergence to be reached. Convergence
in each time step was declared when the residual square root of the sum of the absolute values squared (f-value) met the
absolute convergence tolerance. Based on preliminary model runs, a convergence tolerance of 5 allowed the solver to find
converged solutions in most cases. For each time step, we ran the solver with 100 sets of starting values, except for sensitivity
and model performance evaluations, where we ran 1000 sets of starting values. For each set of starting values, the solver
searched for solutions until convergence was reached, using a maximum of 1500 iterations. For each time step, we identified
the solutions for which the f-value was within the 2.5% quantile, meaning that the 2.5% best solutions were retained.
Subsequently, the average and standard deviation across the 2.5% best solutions for each time step was computed. Our
modeling approach estimates process rates for data sets with a minimum of two time points. Ideally, the time step should be
small enough for steady state conditions (constant process rates) between time points to be supported. In addition, the
difference between open vs. closed system isotope dynamics becomes negligible when small time steps are used, simplifying
model estimates (Denk et al., 2017; Fry, 2006). In our modeling approach, we chose a time step of one day, and assumed open
system isotope dynamics.





**2.6 Calculation of total gross N₂O production and consumption**

PRE estimates gross nitrification derived $N_2O$ production, denitrification derived $N_2O$ production, and $N_2O$ consumption in kg $N_2O$-N ha$^{-1}$ day$^{-1}$ for each depth increment. To calculate the total gross nitrification (total $N_2O_{nit}$) and denitrification derived $N_2O$ (total $N_2O_{den}$) production and gross $N_2O$ consumption (total $N_2O_{consumed}$) for each depth increment over the experimental period, we use trapezoidal integration of daily gross $N_2O$ production and consumption rates. Total gross $N_2O$ production (total $N_2O_{produced}$) for each depth increment is calculated as the sum of total $N_2O_{nit}$ and total $N_2O_{den}$. Finally, total gross $N_2O$

production and consumption rate over the depth profile are determined by summing gross $N_2O$ production and consumption in each depth increment. In addition, we calculated the fraction of total $N_2O$ produced that was emitted as $N_2O$ by dividing cumulative surface fluxes by total $N_2O_{produced}$ ($N_2O$ emitted/gross produced). Alternatively, we defined the fraction of $N_2O$ emitted relative to the $N_2O$ available in the soil profile ($N_2O$ emitted/gross available), by dividing the cumulative surface emission by the sum of total $N_2O$ produced and the initial $N_2O$ in the soil profile at the start of the experiment.

**2.7 Statistical analysis**

The effect of winter wheat variety on cumulative $N_2O$ emissions, $N_2O$ emitted/gross produced and $N_2O$ emitted/gross available was tested using one-way ANOVA for randomized complete block designs. The effect of variety and depth increment on the depth-differentiated total $N_2O_{nit}$, total $N_2O_{den}$, total $N_2O_{produced}$ and total $N_2O_{consumed}$ was tested using a two-way split plot factorial ANOVA with variety as between subject factor, depth as within subject factor and block as a random effect. The

effect of variety, depth and days after transplanting on $N_2O$ concentrations in the soil profile, d$^{15}$N$_{bulk}$, SP and d$^{18}$O was assessed using a three-way mixed effects ANOVA with variety as between subject factor, depth and day after transplanting as within subject factors and block as random effect. For all statistical models, normal distribution of residuals and heterogeneity of variance were evaluated based on QQ-plots, fitted vs. residual plots, Shapiro Wilk test and Levene's test. Appropriate data transformations were implemented in cases where the assumptions were not met. All statistical data analysis was done in R (R

Core Team, 2022). For the three-way mixed effects model, we used packages lmer, predictmeans, and lmerTest, the latter estimating the degrees of freedom of the denominator based on the Satterthwaite approximation.

**3. Results and discussion**

**3.1 N₂O concentrations and isotope values in the soil profile**

Concentrations of $N_2O$ in the soil profile ranged between 0 and 30 ppm (uL $N_2O$ L$^{-1}$), corresponding to a range of 0 to 9.3 g

$N_2O$-N ha$^{-1}$ depth increment$^{-1}$ (Fig. 1). These concentrations are within the range of other studies that measured $N_2O$ concentrations in the soil pore air. For example, van Groenigen *et al.* (2005) observed soil pore air $N_2O$ concentrations between 0.5 and 100 ppm in a sandy excavated peat soil seeded with potatoes (*Solanum tuberosum L.*) in the Netherlands. In a Norway spruce (*Picea abies (L.) Karst.*) on a loamy soil forest in Germany, soil pore air $N_2O$ concentrations typically remained below 10 ppm, except for plots that experienced impacts of freeze-thaw cycles following snow removal, where concentrations spiked





up to 50 ppm (Goldberg et al., 2010). In a monolith lysimeter experiment on a loamy soil in Michigan, USA, mean subsurface $N_2O$ concentrations over the growing season stayed below 8 ppm in annual cropping systems, and below 3 ppm in perennial crops (Shcherbak and Robertson, 2019). In our study, the highest $N_2O$ concentrations were observed in the 45-75cm and 75-105 cm depth increment. This is similar to other studies where $N_2O$ concentrations where typically greater at deeper compared to shallower soil depths (Shcherbak and Robertson, 2019; Goldberg et al., 2010; Van Groenigen et al., 2005). Subsurface

concentrations at depths of 50-150 cm observed in the literature and shown in this study are commonly 20 to 100 times greater than atmospheric $N_2O$ concentrations, underlining the importance of understanding the role of subsurface $N_2O$ dynamics in driving surface fluxes and ecosystem N loss through complete denitrification. Subsurface concentrations changed slowly and steadily over time, which is in stark contrast to the typical short-lived $N_2O$ pulses observed in surface fluxes (Decock et al., 2017; Verhoeven et al., 2017). Similarly, subsurface isotope values of $N_2O$ showed a relatively consistent temporal pattern

(Fig. 1). The temporal patterns in subsurface $N_2O$ concentrations and isotope concentrations observed in this and other studies (van Groenigen 2005, Verhoeven 2019) suggest that a sampling frequency of once every two weeks or once a month may be sufficient to adequately capture subsurface $N_2O$ dynamics.

    We observed bulk $d^{15}N$ isotope values of $N_2O$ between -17.6 and 11.9‰, $d^{18}O$ isotope values of $N_2O$ between 22.6 and 59.9‰, and SP values of -33.3 to 35.7‰ (Fig. 1). Isotope values for $d^{15}N$ and $d^{18}O$ of $N_2O$ were in the range of other studies that

measured $N_2O$ isotope values in the depth profile (Yano et al., 2014; Verhoeven et al., 2019; Zou et al., 2014; Koehler et al., 2012). With respect to SP, some of our measurements show lower values than typically observed in the literature, where values tend to range between -15 and 60‰ (Decock and Six, 2013c). However, out of 727 observations, only 8 values were below -15‰, indicating that our measurements are generally in line with literature observations. $d^{15}N$ values were lower in deeper compared to shallower soil increments, especially early in the growing season. Similarly, van Groenigen et al. (2005) observed

lower $d^{15}N$ values at deeper compared to shallower depth, which was attributed to production of $N_2O$ in the subsoil. In contrast, $d^{15}N$ of $N_2O$ was commonly lower and $N_2O$ concentrations greater at 5 cm compared to 12.5 and 25 cm depth in a paddy rice cropping system in Italy (Verhoeven et al. 2019), indicating that the depth of $N_2O$ production and $N_2O$ dynamics over the soil profile vary between cropping systems, as well as the depths considered in a given study. Both $d^{15}N$ and $d^{18}O$ values tended to steadily increase for all soil depths and varieties over the duration of the experiment, while this pattern existed in a less

pronounced way for SP. Moreover, $N_2O$ concentrations in the soil profile were greater at the start compared to the end of our experiment across all soil columns. Despite the general trends in temporal patterns, there was a significant depth by variety interactive effect on $N_2O$ concentrations and isotope values (Table 2), suggesting differences in $N_2O$ dynamics between varieties. The increase of multiple isotope values of $N_2O$, as observed in this study, are commonly associated with $N_2O$ reduction to $N_2$ (Van Groenigen et al., 2005; Jinuntuya-Nortman et al., 2008; Well and Flessa, 2009). Any change in isotope

values of $N_2O$, however, can also be affected by a shift in the relative contribution of $N_2O$ source processes or a change in isotope values of precursors (Decock and Six, 2013b). Alternatively, a net decrease in $N_2O$ concentrations in the soil profile could be driven by upward movement and losses of $N_2O$ to surface fluxes, thus soil profile $N_2O$ concentrations may reflect a





combination of these processes. This highlights the need for a new modeling approach to quantify the contribution of different source processes to subsurface $N_2O$ dynamics.

**3.2 Model performance**

The first steps in modeling $N_2O$ production and consumption include calculating diffusion fluxes for each depth increment and fitting smooth lines to the temporal patterns of the input variables. In this study, diffusion fluxes are estimated based on Fick's law, with the gas diffusion coefficient based on the assumption that $N_2O$ diffuses through both air and water in the soil pore space (Millington and Quirk, 1960; Mccarthy and Johnson, 1995; Yano et al., 2014). An elaborate discussion on strenghts and limitations of approaches to estimate diffusion is beyond the scope of this study. However, the estimated diffusivity is reasonable for the repacked soil column where homogenized soils are expected. Also, it should be noted that predicting the movement of $N_2O$ in soil goes beyond diffusion, as it is affected by phenomena such as entrapment, and convection (Clough et al., 2005). Moreover, aggregated soils are characterized by inter- and intra-aggregate pore spaces, further complicating predictions of how gases such as $N_2O$ move through the soil (Jayarathne et al., 2020). As research on diffusion and other movement of $N_2O$ in soil continuously advances, there is opportunity for future improvements of the gas diffusion module in PRE.

Curve fitting of temporal patterns in the input variables $N_2O$ concentrations, SP, $d^{18}O$ of $N_2O$, diffusion influxes from the bottom and top of the depth increment, and outflux of $N_2O$ from the depth increment through diffusion is illustrated for the 45-75cm depth increment for one of the columns (Fig. 2). In the example provided, $N_2O$ concentrations and diffusion fluxes are less variable and show a cleaner temporal pattern compared SP and $d^{18}O$ of $N_2O$. This finding was common across columns and depth increments. Nevertheless, for each depth increment in each column, we were able to distinguish clear temporal patterns in all input variables. Success of the modelling strategy presented in this study depends on a sufficiently high sampling frequency, precise and accurate isotope analyses, and reliable bulk density and soil moisture measurements.

An important source of potential uncertainty in any model is uncertainty introduced by fixed model parameters and input variables. In an assessment of the dual isotope mapping appraoch proposed by Lewicka-Szcebak *et al.* (2017), it was suggested that using a study-specific instead of default isotope endmember values could largely improve model performance (Wu et al., 2019). Given that the modeling approach to estimate $N_2O$ production and consumption presented in this study heavily relies on the accuracy of input variables and literature values on isotope effects, it is imperative that the robustness of model predictions in response to uncertainty around input variables and isotope effects is assessed. To this end, we assessed how estimates of $N_2O_{nit}$, $N_2O_{den}$ and $N_2O_{red}$ are affected by randomly selecting isotope effects from a normal distribution or range of values reported in the literature (Table 1, Fig. 3b). Likewise, we compared how PRE constrains estimates of $N_2O_{nit}$, $N_2O_{den}$ and $N_2O_{red}$ when input variables are drawn randomly from a normal distribution around input variables in each iteration compared to when average values of input variables in each time point were used (Fig. 3c). We observed similar patterns in $N_2O_{nit}$, $N_2O_{den}$ and $N_2O_{red}$ over time when uncertainty in input variables and isotope effects were introduced, compared to when input variables and isotope effects were fixed in each time point. Moreover, differences in temporal patterns of $N_2O_{nit}$, $N_2O_{den}$





and $N_2O_{red}$ between depths and varieties were much greater compared to differences associated with introducing uncertainty from input variables and isotope effects (Fig. 3 and 4). The results suggest that PRE is robust in demonstrating treatment effects when constraining $N_2O$ production and consumption rates over the depth profile.

### 3.3 Gross $N_2O$ production, consumption, and $N_2O$:($N_2O$+$N_2$) ratios

Gross $N_2O$ production across the soil profile ranged between 3.9 and 152.5 kg N ha$^{-1}$ over the experimental period (144 days), while $N_2O$ consumption ranged between 4.1 and 170.3 kg N ha$^{-1}$ (Fig. 5). In comparison, data for total denitrification reported in the literature is poorly constrained and varies widely across studies, in part due to methodological limitations in quantifying total denitrification rates. In a global synthesis, denitrification rates, defined as $N_2O$+$N_2$ fluxes, were found to range between 0.001 and 20 kg N ha$^{-1}$ day$^{-1}$ (Pan et al., 2022). When calculating mean daily gross $N_2O$ production in our 144-day experiment,

we find rates between 0.03 and 1.06 kg N ha$^{-1}$ day$^{-1}$, well within but on the low end of the range observed in the literature. Data on denitrification rates reported in the literature is typically derived from laboratory incubation experiments, using inhibitors, $^{15}N$ tracers, or controlled systems with an $N_2$ free atmosphere (Friedl et al., 2020). Disturbance associated with these experimental setups may enhance denitrification, thereby inflating estimates on terrestrial ecosystem $N_2$ loss. On the other hand, denitrification rates determined using isotope pool dilution were shown to be drastically underestimated compared to

rates determined by the gas-flow soil core method, a method that directly measures gross $N_2O$ production and consumption in soil by simultaneously quantifying $N_2O$ and $N_2$ fluxes in an $N_2$-free controlled environment, without the use of an inhibitor or $^{15}N$ labelling of substrate (Wen et al., 2016). Thus, experimental conditions can greatly bias the quantification of total denitrification rates, which highlights the importance of methods such as the one presented here, providing a low disturbance, field deployable option for quantification of denitrification *in situ*.

An indirect way to evaluate the range of denitrification N loss in an ecosystem is by using an N budgeting approach. In our study, fertilizer N was applied at a rate of 140 kg N ha$^{-1}$. In addition, initial soil $NO_3^-$ concentrations at the beginning of the experiment in the 1.35 m depth profile were relatively high, totaling 199-336 kg N ha$^{-1}$. In relation to the potentially available N from fertilizer and initial soil $NO_3^-$, the denitrification loss over the course of the experiment in our study ranged between 1 % and 35 %. Similarly, Pan et al. (2022) found that fertilizer N losses from denitrification across studies ranged between 0.5

% and 40%. This range aligns well with findings from N budget studies using $^{15}N$ tracers, where the percentage of N that is unaccounted for was on average 38% (Gardner and Drinkwater, 2009). Unaccounted N in $^{15}N$ budget studies is typically attributed to N losses in the form of leaching, volatilization and denitrification. Which loss pathway dominates is important, however, in mitigating N pollution. Our method can help guide environmental regulation and the development of mitigation strategies for N pollution, by better resolving the contribution of total denitrification versus more harmful loss pathways to

total N loss.

Besides evaluating gross $N_2O$ production and consumption in the context of ecosystem N budgets, gross $N_2O$ production and consumption rates can be contextualized in relation to other N transformation processes including gross mineralization, nitrification and immobilization. Considering an experimental period of 144 days, a soil depth of 1.35 m and an average bulk





density of 1.7 g cm$^{-3}$, gross N$_2$O production and consumption in our study ranged between 1.2 and 51.5 mg N kg$^{-1}$ soil day$^{-1}$.
This is well within the range of other key N transformation processes, namely, gross mineralization, nitrification and immobilization, which were found to range between 0.1 and 100 mg N kg$^{-1}$ soil day$^{-1}$ across ecosystem types (Booth et al., 2005). It should not come as a surprise that gross N$_2$O production and consumption rates are in the same range as other N transforming processes, given the increasing evidence from molecular techniques for high abundance and diversity of denitrifying organisms in the soil (Hallin et al., 2018; Wei et al., 2015). In contrast, surface N$_2$O emissions commonly range
in the order of 0.1 to 100 ug N kg$^{-1}$ soil day$^{-1}$ (Decock and Six, 2013a), two to three orders of magnitude smaller than the processes driving N$_2$O emissions. This implies that if we want to manage these small surface fluxes of N$_2$O, we are fighting against a large background of potential N$_2$O producing and consuming processes both at deep and shallow depths in the soil profile.

Across all varieties, mean N$_2$O consumption rates were slightly greater than mean N$_2$O production rates (Fig. 5), suggesting
that N$_2$O consumption drove the net decrease in soil N$_2$O content observed at the end compared to the beginning of the experiment. As such, the soil in this experiment behaved, to some extent, as a sink of N$_2$O. While it is generally accepted that soils are net sources of N$_2$O, it is well known that soil can act as a sink under certain conditions (Chapuis-Lardy et al., 2007; Liu et al., 2022). Main direct controls on N$_2$O consumption include microbial community structure, soil oxygen concentrations, forms and concentrations of N sources, residue or other amendments, soil pH, and copper and iodide concentrations (Liu et al.
2022). In our study, cumulative N$_2$O surface emissions were small but positive and ranged between 0.10 and 0.54 kg N ha$^{-1}$ over the experimental period (Table 5), indicating that despite some net sink behavior in the soil profile, the soil surface in our experiment was a net source of N$_2$O.

Given the difficulties in quantifying N$_2$ fluxes, several studies have used ratios of N$_2$O:(N$_2$O+N$_2$) determined under laboratory conditions in combination with field-scale N$_2$O emissions data to estimate N$_2$ loss at the field, ecosystem or global scale
(Schlesinger, 2009; Scheer et al., 2020; Wang et al., 2020). Synthesizing 39 studies, including laboratory and field experiments across various ecosystems, Schlesinger et al. (2009) found an average N$_2$O:(N$_2$O+N$_2$) ratio of 0.082 ± 0.024. However, N$_2$O:(N$_2$O+N$_2$) ratios were highly variable across studies, ranging between 0 and 1. Scheer *et al.* (2020) suggested mean N$_2$O:(N$_2$O+N$_2$) ratios of 0.109 ± 0.020 for agricultural soils, 0.124 ± 0.031 for soils under natural vegetation, and 0.020 ± 0.009 for freshwater wetlands and flooded soils. Meanwhile, Wang et al. (2020) observed significant temporal variation in
N$_2$O:(N$_2$O+N$_2$), ranging from 0.1 to 0.7 over the course of their experiment, suggesting that temporally explicit N$_2$O:(N$_2$O+N$_2$) ratios are essential to constrain field-scale N$_2$ losses. It should be noted that across these studies, ratios are generally determined under experimental conditions that only capture N$_2$O and N$_2$ production in the topsoil. When dividing cumulative surface N$_2$O emissions observed in our study by the total gross N$_2$O production over the depth profile, N$_2$O:(N$_2$O+N$_2$) ratios range between 0.003 and 0.052 or 0.3 % and 5 % (Table 5), much lower than mean values for agricultural soils reported in the literature. The
ratios found in our study are even lower, when calculated by dividing cumulative surface N$_2$O emissions by gross N$_2$O production plus initial N$_2$O in the soil profile, where the denominator represents N$_2$O potentially available for consumption





(Table 5). This suggests that any approach to estimate denitrification that only considers topsoil N dynamics likely underestimates total ecosystem $N_2$ loss.

### 3.4 Case study: Effect of wheat variety on gross $N_2O$ production and consumption

In our study, there was a significant effect of variety on gross $N_2O$ production and consumption (Table 4). The older variety Mont Calme 268 showed significantly greater gross $N_2O$ production compared to the newer variety Zinal, while $N_2O$ consumption was significantly greater in both older varieties compared to Zinal (Table 3). In the same experiment, it was shown earlier that the root biomass was significantly greater in the older varieties compared to the newer variety Zinal (Van De Broek et al., 2020). The other newer variety, CH Claro, showed no significant differences in root biomass, $N_2O$ production

or $N_2O$ consumption with either the other newer variety or the older varieties. These observations partially corroborate our hypothesis that increased root biomass would increase gross $N_2O$ production and consumption. Our results are in line with many other studies that have observed increased denitrification with increased belowground carbon inputs from roots or root exudates (Wang et al., 2021; Malique et al., 2019; Klemedtsson et al., 1987; Qian et al., 1997).

The main knowledge gap, however, consists of not knowing the amount of reactive N that can be removed through

denitrification under field conditions, and understanding how plants affect $N_2O$ emission by shifting the balance between $N_2O$ production and consumption. In our study, cumulative $N_2O$ emissions were significantly greater in Mont Calme 268 compared to CH Claro, but similar to $N_2O$ emissions from Probus and Zinal (Table 5). The difference in effects of variety on gross $N_2O$ production and consumption across the soil profile versus surface emissions suggests that subsurface $N_2O$ dynamics may be partially decoupled from surface $N_2O$ emissions. This is in line with studies comparing $N_2O$ surface fluxes based on chamber

measurements with the soil gradient method, which have suggested that a mismatch in $N_2O$ flux estimates between the two methods is due to the inability to adequately capture $N_2O$ dynamics in the 10-20 cm of soil (Wolf et al., 2011; Yao et al., 2018). This partial decoupling of subsurface and surface $N_2O$ dynamics implies that studies pertaining to the effect of management on soil denitrification may need to consider whether the primary goal is to minimize $N_2O$ emissions or to remove reactive N from the soil profile, or both, and optimize the experimental design accordingly.

Ratios of $N_2O:(N_2O+N_2)$, defined as $N_2O$ emitted over gross $N_2O$ produced or available in our study, were not different between varieties, indicating that wheat varieties did not shift the balance between $N_2O$ production and consumption. Across the literature, observations of the impact of plants and belowground carbon inputs on $N_2O$ emissions and $N_2O:(N_2O+N_2)$ are much more divergent compared to observation on potential denitrification. In a mesocosm experiment, barley plants decreased $N_2O$ emissions compared to unplanted controls, while the abundance of genes involved in denitrification was increased (Wang

et al., 2021). In contrast, wheat plants stimulated both $N_2O$ emissions and denitrification compared to an unplanted control in a similar mesocosm experiment (Ai et al., 2020). It has been suggested that root-derived C may stimulate denitrification and $N_2O$ emissions only when soil $NO_3^-$ is not limited and $O_2$ concentrations are low (Rummel et al., 2021). In the latter study, $N_2O:(N_2O+N_2)$ ratios were not affected by fertilizer treatment or plant type (Rummel et al., 2021). Overall, denitrification and $N_2O:(N_2O+N_2)$ ratios have been shown to be affected by crop type, the quality of organic matter inputs, as well as soil type





and environmental conditions (Malique et al., 2019; Henry et al., 2008; Wang et al., 2020). Our novel approach, capable of quantifying gross $N_2O$ production and consumption rates *in situ*, can aid future research aimed at closing knowledge gaps regarding the impact of plant traits, management, and environmental conditions on denitrification as it pertains to $N_2O$ emissions and removal of reactive N from the soil profile.

## 4. Conclusion

This study aimed to provide proof-of-concept for a novel approach to estimate the rates of gross $N_2O$ consumption or total denitrification in the soil profile, by combining $N_2O$ diffusion with isotope mixing and fractionation models. The model is referred to as Process Rate Estimator or PRE. We tested PRE in a greenhouse lysimeter experiment, assessing the impact of wheat varieties known to have different root biomass on gross $N_2O$ production and consumption. PRE was able to constrain daily gross nitrification and denitrification derived $N_2O$ production rates, as well as gross $N_2O$ reduction to $N_2$, across the depth

profile. Variability in model estimates due to uncertainty around input variables and isotope end-members was minor compared to variation in $N_2O$ production and consumption rates between depth increments and wheat varieties, demonstrating that PRE is robust in assessing impacts of management or environmental drivers on total soil denitrification. Gross $N_2O$ production and consumption ranged between 3.9 and 170.3 kg N ha$^{-1}$ over the experimental period, illustrating the importance of total denitrification in ecosystem N budgets. Gross production and consumption of $N_2O$ peaked in the 0-15cm and 15-45cm depth

increments, while $N_2O$ concentrations reached maxima in the 45-75cm and 75-105cm depth increments. As such, identifying the depth of maximum $N_2O$ accumulation was instrumental in constraining diffusion fluxes and estimating total denitrification across the soil profile. Our study demonstrates that by using the intermediate denitrification product, $N_2O$, in combined diffusion and isotope mixing and fractionation models, total denitrification over the soil profile can be constrained. We conclude that our method opens new opportunities to potentially identify management strategies that can curtail pollution

associated with the dispersion of reactive N in terrestrial ecosystems.

## Appendix A: Proof of state function for infinitesimal changes in isotope value over time

For simplicity, the proof of equations used to describe the change of isotope values over time is described for the scenario where there are two incoming processes ($k_{in,1}$ and $k_{in,2}$) mixing with a pool of known concentration and isotope value and two

outgoing processes ($k_{out,1}$ and $k_{out,2}$) that induce fractionation isotope effects. In this scenario, the change in isotope value over time ($DI/dt$) can be described by eq. A.1, using a combination of mixing and fractionation (Fry, 2006):

Equation A.1:

$$\frac{\Delta I}{\Delta t} = \frac{I_t - I_0}{\Delta t} =$$






$$\left(\left(\frac{\begin{array}{c}\dfrac{I_0 C_0 + I_{in,1}k_{in,1}\Delta t + I_{in,2}k_{in,2}\Delta t}{C_0 + k_{in,1}\Delta t + k_{in,2}\Delta t}\\[6pt]-\eta_{out,1}\left(1 - \dfrac{C_0 + k_{in,1}\Delta t + k_{in,2}\Delta t - k_{out,1}\Delta t}{C_0 + k_{in,1}\Delta t + k_{in,2}\Delta t}\right)\\[6pt]-\eta_{out,2}\left(1 - \dfrac{C_0 + k_{in,1}\Delta t + k_{in,2}\Delta t - k_{out,2}\Delta t}{C_0 + k_{in,1}\Delta t + k_{in,2}\Delta t}\right) - I_0\end{array}}{}\right)\right) \Big/ \Delta t$$

Where $I_0$ and $I_t$ refer to the isotope value at the beginning and end of each time step, respectively; $Dt$ is the length of the time step; $C_0$ is the concentration of the pool for which the change in isotope value is described at the beginning of the time step; $k_{in,1}$ and $k_{in,2}$ represent the rates at which new product from process 1 and 2 is coming into the pool for which the change in isotope value is described; $I_{in,1}$ and $I_{in,2}$ are the isotope values associated with new product coming in from process 1 and 2, respectively; $k_{out,1}$ and $k_{out,2}$ represent process 1 and 2 by which product is leaving the pool for which the change in isotope value is described; and $\eta_{out,1}$ and $\eta_{out,2}$ are the isotope effect associated with process 1 and process 2. The equation can be rearranged as follows:

Equation A.2:

$$\frac{\Delta I}{\Delta t} = \big(I_0 C_0 + I_{in,1}k_{in1}\Delta t + I_{in,2}k_{in,2}\Delta t - \eta_{out,1}(C_0 + k_{in,1}\Delta t + k_{in,2}\Delta t - C_0 - k_{in,1}\Delta t - k_{in,2}\Delta t + k_{out,1}\Delta t) - \eta_{out,2}(C_0$$
$$+ k_{in,1}\Delta t + k_{in,2}\Delta t - C_0 - k_{in,1}\Delta t - k_{in,2}\Delta t + k_{out,2}\Delta t) - I_0 C_0 - I_0 k_{in,1}\Delta t - I_0 k_{in,2}\Delta t)\big)$$
$$/((C_0 + k_{in1}\Delta t + k_{in2}\Delta t)\Delta t)$$

With various terms cancelling out, the equation can be rewritten as:

Equation A.3:

$$\frac{\Delta I}{\Delta t} = \frac{(I_{in,1}k_{in,1}\Delta t + I_{in,2}k_{in,2}\Delta t - \eta_{out,1}k_{out,1}\Delta t - \eta_{out,2}k_{out,2}\Delta t - I_0 k_{in,1}\Delta t - I_0 k_{in,2}\Delta t)}{(C_0 + k_{in,1}\Delta t + k_{in,2}\Delta t)\Delta t}$$

After cancelling out Dt in denominator and enumerator terms where appropriate, the equation simplifies to:

Equation A.4:

$$\frac{\Delta I}{\Delta t} = \frac{(I_{in,1}k_{in,1} + I_{in,2}k_{in,2} - \eta_{out,1}k_{out,1} - \eta_{out,2}k_{out,2} - I_0 k_{in,1} - I_0 k_{in,2})}{(C_0 + k_{in,1}\Delta t + k_{in,2}\Delta t)}$$

For $\Delta t$ approaching 0, the equation can be further simplified as:

Equation A.5:

$$\frac{\Delta I}{\Delta t} = \frac{k_{in,1}(I_{in,1} - I_0) + k_{in,2}(I_{in,2} - I_0) - \eta_{out,1}k_{out,1} - \eta_{out,2}k_{out,2}}{C_0}$$





**Generalized:**

Equation A.6:

$$\frac{\Delta I}{\Delta t} = \frac{\sum_{i=1}^{n} k_{in,i}(I_{in,i} - I_0) - \sum_{j=1}^{m} \eta_{out,j} k_{out,j}}{C_0}$$


Note that this generalized model is a linear model, with $k_{in,i}$ and $k_{out,j}$ as unknowns. By linearilizing the equations, solutions to the system of equations can be found more easily.

**Code available:** When accepted for publication, code will be made available on the author's website and a link will be provided

**Author contribution**

CD led the investigation, including acquisition of funding, development of methodology and model, formal analysis, and writing of the original draft of the manuscript. JS and JL contributed to funding acquisition. JS and FC advised on project conceptualization, design, and execution. JS and MB advised on lysimeter design and construction. JL advised on model design and assisted in writing the model code. MB took a lead on stable isotope analysis and provided technical assistance for the experimental design and implementation. EV assisted in model conceptualization and design. All authors contributed to the

writing of the manuscript.

**Competing interests**

The authors declare that they have no conflict of interest.

**Acknowledgements**

We thank Benjamin Wilde and Chris Mikita for technical support in design and construction of lysimeters and assistance in

gas sampling. We further thank Brigitta Herzog and Hansueli Zellweger for their help with greenhouse management and plant protection at the ETH Research Station for Plant Sciences in Lindau.

This research has been supported by Plant Fellows, a postdoctoral fellowship administered by the Zurich Basel Plant Science Center and funded under the European Union's Seventh Framework Programme for research, technological development and demonstration (grant no. GA-2010-267243 – Plant Fellows); the Swiss National Science Foundation (project numbers

205321_153545 "CarIN" and 200021_160232); and ETH core start-up funds provided to Johan Six.





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

**Table 1:** Isotope endmembers values used for modeling gross nitrification derived $N_2O$ production, gross denitrification derived $N_2O$
production, and gross $N_2O$ reduction.

| End-member | Value (‰) | General model runs References | Value (‰) | Uncertainty analysis** References |
|---|---|---|---|---|
| $\eta_{SP,dif}$ | 1.55 | Well and Flessa, 2008 | 1.55±0.28 | Well and Flessa, 2008 |
| $\eta_{18O,dif}$ | -7.79 | Well and Flessa, 2008 | -7.79±0.27 | Well and Flessa, 2008 |
| $SP_{nit}$ | 34.4 | Decock and Six 2013b | 26.2 to 34.6 | Denk et al. 2017 |
| $\delta^{18}O_{nit}$ | 36.5 | Lewicka-Szczebak *et al.* 2017 | 36.5±2 | Arbitrary standard deviation |
| $SP_{den}$ | -2.4 | Decock and Six 2013b | -2.4 to -0.9 | Denk et al. 2017 |
| $\delta^{18}O_{den}$ | 11.1 | Lewicka-Szczebak *et al.* 2017* | 11.1±2 | Arbitrary standard deviation |
| $\eta_{SP,red}$ | -5.3 | Denk *et al.* 2017 | -8 to -2 | Denk et al. 2017 |
| $\eta_{18O,red}$ | -16.1 | Lewicka-Szczebak *et al.* 2017, based on a ratio of $\eta_{SP,red}/\eta_{18O,red}$ of 0.33 | Variable | Arbitrary standard deviation of 0.05 for $\eta_{SP,red}/\eta_{18O,red}$ of 0.33 |

*Lewicka-Szczebak *et al.* (2017) originally report $\delta^{18}O\text{-}N_2O(N_2O/H_2O)$. Thus, to calculate a pure $\delta0^{18}O\text{-}N_2O$, we added the $\delta^{18}O\text{-}H_2O$ value used in our study (-9.9‰). **When a range is indicated, a random number from the range was selected in each iteration. When an average and standard deviation is indicated, a random value was drawn from the normal distribution in each iteration.




**Table 2:** F and p values for analysis of variance testing the effect of depth, variety and time (day) on $N_2O$ concentrations, $\delta^{15}Nbulk$, SP, and $\delta^{18}O$.

|  | $N_2O$ concentration | | $\delta^{15}Nbulk$ | | SP | | $\delta^{18}O$ | |
|---|---|---|---|---|---|---|---|---|
|  | F | p | F | p | F | p | F | p |
| variety | 23.1 | <0.001 | 4.4 | 0.005 | 3.1 | 0.025 | 17.7 | <0.001 |
| depth | 19.9 | <0.001 | 11.7 | 0.009 | 18.7 | <0.001 | 13.9 | <0.001 |
| day | 2.1 | 0.007 | 5.7 | <0.001 | 16.3 | <0.001 | 5.1 | <0.001 |
| variety:depth | 4.5 | <0.001 | 2.3 | 0.009 | 1.7 | 0.058 | 4.0 | <0.001 |
| variety:day | 0.7 | 0.917 | 0.8 | 0.815 | 1.8 | 0.001 | 1.1 | 0.378 |
| depth:day | 0.4 | 1.000 | 0.7 | 0.939 | 0.6 | 0.997 | 0.9 | 0.601 |
| variety:depth:day | 0.2 | 1.000 | 0.4 | 1.000 | 0.5 | 1.000 | 0.5 | 1.000 |






**Table 3:** Cumulative nitrification ($N_2O_{nit}$) and denitrification ($N_2O_{den}$) derived $N_2O$ production, cumulative gross $N_2O$ production ($N_2O$ produced) and cumulative gross $N_2O$ consumption ($N_2O$ consumed, here considered total denitrification) by depth and variety (Mean ± standard errors, in kg N ha$^{-1}$ for each depth increment). Different letters indicate significant differences between depths or varieties.

| | $N_2O_{nit}$ | | $N_2O_{den}$ | | $N_2O$ produced | | $N_2O$ consumed | |
|---|---|---|---|---|---|---|---|---|
| | kg N ha$^{-1}$ depth increment$^{-1}$ | | | | | | | |
| *Mont Calme 268* | | | | | | | | |
| 7.5 | 18.94 ± 30.63 | | 30.11 ± 47.47 | | 49.05 ± 78.1 | | 56.49 ± 89.6 | |
| 30 | 3.08 ± 2.24 | | 8.62 ± 8.84 | | 11.7 ± 10.96 | | 12.57 ± 14.8 | |
| 60 | 1.01 ± 0.64 | | 1.45 ± 0.77 | | 2.46 ± 1.28 | | 1.68 ± 0.39 | |
| 90 | 0.48 ± 0.37 | | 0.78 ± 0.49 | | 1.26 ± 0.86 | | 2.64 ± 2.1 | |
| 120 | 0.05 ± 0.02 | | 0.13 ± 0.03 | | 0.18 ± 0.03 | | 0.36 ± 0.05 | |
| *Probus* | | | | | | | | |
| 7.5 | 3.05 ± 2.94 | | 6.64 ± 5.7 | | 9.69 ± 8.62 | | 12.25 ± 10.97 | |
| 30 | 2.15 ± 0.59 | | 12 ± 12.86 | | 14.15 ± 13.17 | | 15.43 ± 14.85 | |
| 60 | 0.74 ± 0.73 | | 2.76 ± 4.31 | | 3.5 ± 5.02 | | 3.53 ± 3.73 | |
| 90 | 0.42 ± 0.63 | | 0.2 ± 0.11 | | 0.62 ± 0.67 | | 2.18 ± 2.99 | |
| 120 | 0.16 ± 0.26 | | 0.36 ± 0.39 | | 0.52 ± 0.64 | | 1.05 ± 1.32 | |
| *Zinal* | | | | | | | | |
| 7.5 | 5.5 ± 4.73 | | 6.08 ± 6.17 | | 11.58 ± 10.78 | | 12.61 ± 12.83 | |
| 30 | 0.98 ± 0.33 | | 3.86 ± 2.29 | | 4.83 ± 2.25 | | 5.23 ± 1.97 | |
| 60 | 0.32 ± 0.21 | | 0.41 ± 0.48 | | 0.73 ± 0.68 | | 1.35 ± 1.3 | |
| 90 | 0.05 ± 0.04 | | 0.11 ± 0.06 | | 0.16 ± 0.09 | | 0.17 ± 0.07 | |
| 120 | 0.05 ± 0.04 | | 0.13 ± 0.12 | | 0.17 ± 0.15 | | 0.26 ± 0.18 | |
| *CH Claro* | | | | | | | | |
| 7.5 | 0.86 ± 0.24 | | 1.84 ± 0.95 | | 2.7 ± 1.16 | | 3.05 ± 1.58 | |
| 30 | 0.86 ± 0.44 | | 1.87 ± 1.26 | | 2.72 ± 1.69 | | 2.98 ± 2.23 | |
| 60 | 0.15 ± 0.07 | | 0.84 ± 0.77 | | 0.99 ± 0.83 | | 1.04 ± 0.69 | |
| 90 | 0.11 ± 0.04 | | 0.36 ± 0.12 | | 0.47 ± 0.08 | | 0.69 ± 0.11 | |
| 120 | 0.05 ± 0.03 | | 0.3 ± 0.1 | | 0.35 ± 0.11 | | 0.55 ± 0.15 | |
| *Means by depth across varieties** | | | | | | | | |
| 7.5 | 7.09 ± 15.17 | a | 11.17 ± 23.6 | a | 18.26 ± 38.74 | a | 21.1 ± 44.53 | a |
| 30 | 1.77 ± 1.39 | a | 6.59 ± 7.92 | a | 8.35 ± 8.89 | a | 9.05 ± 10.49 | a |
| 60 | 0.55 ± 0.55 | b | 1.37 ± 2.12 | b | 1.92 ± 2.55 | b | 1.9 ± 2 | b |
| 90 | 0.27 ± 0.37 | bc | 0.36 ± 0.35 | bc | 0.63 ± 0.63 | bc | 1.42 ± 1.89 | b |
| 120 | 0.08 ± 0.12 | c | 0.23 ± 0.21 | c | 0.31 ± 0.32 | c | 0.56 ± 0.65 | b |
| *Means by variety across depths** | | | | | | | | |
| Mont Calme 268 | 4.71 ± 13.79 | a | 8.22 ± 21.72 | a | 12.93 ± 35.44 | a | 14.75 ± 40.81 | a |
| Probus | 1.3 ± 1.66 | ab | 4.39 ± 7.23 | ab | 5.7 ± 8.38 | ab | 6.89 ± 9.41 | a |
| Zinal | 1.38 ± 2.81 | b | 2.12 ± 3.55 | b | 3.5 ± 6.18 | b | 3.92 ± 6.94 | b |
| CH Claro | 0.41 ± 0.43 | b | 1.04 ± 0.98 | ab | 1.45 ± 1.38 | ab | 1.66 ± 1.57 | ab |

*Letter codes are based Tukey pairwise comparisons of log transformed data






**Table 4:** F and p values for analysis of variance testing the effect of depth and variety on nitrification ($N_2O_{nit}$) and denitrification ($N_2O_{den}$) derived $N_2O$ production, total $N_2O$ production ($N_2O$ produced) and total $N_2O$ consumption ($N_2O$ consumed).

|  | $N_2O_{nit}$ | | $N_2O_{den}$ | | $N_2O$ produced | | $N_2O$ consumed | |
|---|---|---|---|---|---|---|---|---|
|  | F | p | F | p | F | p | F | p |
| variety | 4.07 | 0.013 | 3.40 | 0.027 | 3.93 | 0.016 | 4.24 | 0.011 |
| depth | 32.57 | <0.001 | 27.40 | <0.001 | 31.86 | <0.001 | 17.14 | <0.001 |
| variety*depth | 0.57 | 0.850 | 1.02 | 0.450 | 0.88 | 0.570 | 0.74 | 0.701 |


**Table 5.** Cumulative surface $N_2O$ emissions (kg N ha$^{-1}$), the fraction of gross produced $N_2O$ that was emitted as $N_2O$ (%) and the fraction of gross available $N_2O$ (gross production plus initial soil $N_2O$ content) that was emitted as $N_2O$ (%) for each variety. Lower case letters indicate significant differences between varieties.

| *Variety* | Cumulative $N_2O$ kg N ha$^{-1}$ | | $N_2O$ emitted/ gross produced % | | $N_2O$ emitted/ gross available % | |
|---|---|---|---|---|---|---|
| Mont Calme 268 | 0.41±0.02 | a | 2.2±1.5 | a | 1.4±0.7 | a |
| Probus | 0.19±0.01 | ab | 1.1±0.4 | a | 0.9±0.3 | a |
| Zinal | 0.23±0.02 | ab | 2.3±1.3 | a | 1.6±0.7 | a |
| CH Claro | 0.13±0.01 | b | 1.9±0.1 | a | 1.3±0.2 | a |



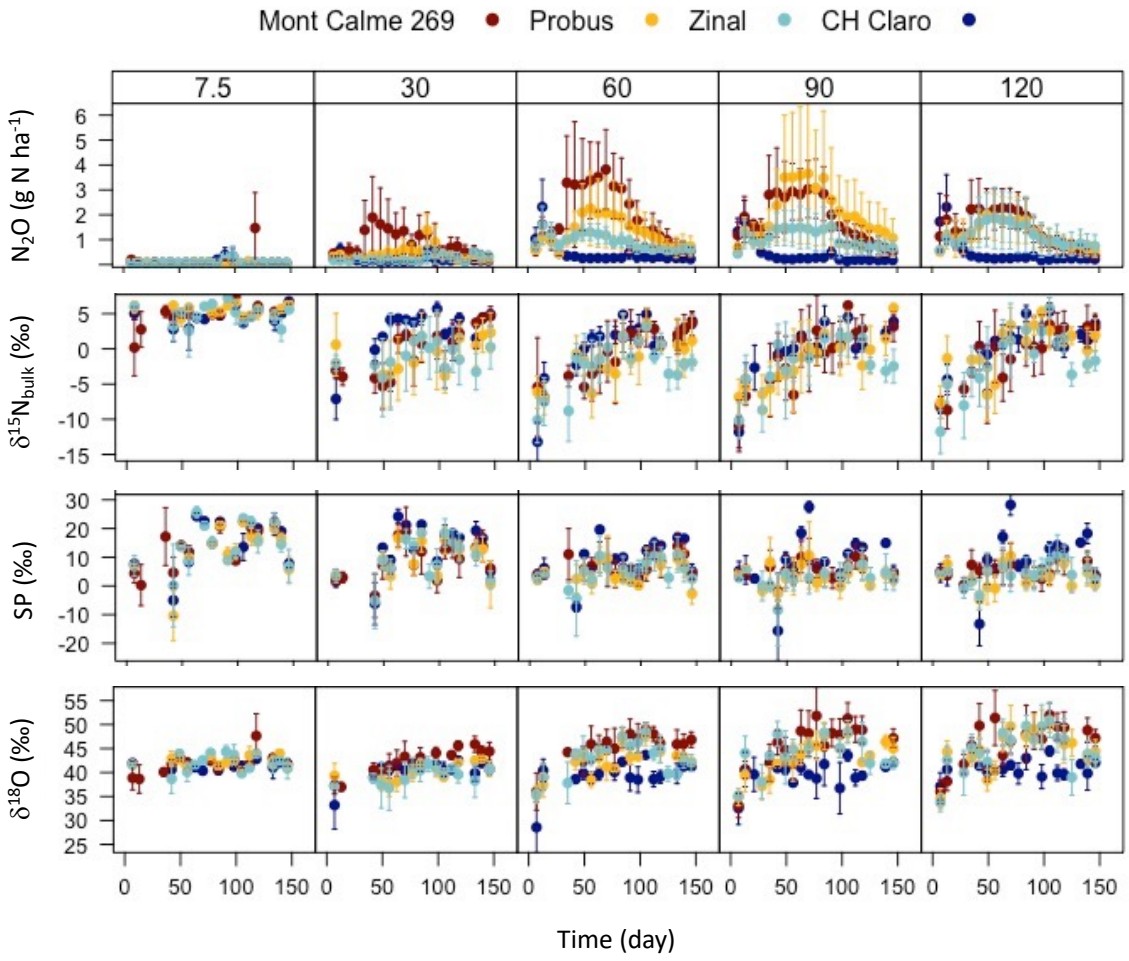

**Figure 1:** Nitrous oxide concentrations (g $N_2O$-N ha$^{-1}$), $\delta^{15}N_{bulk}$ (‰), SP (‰), and $\delta^{18}O$ (‰) of $N_2O$ for the four wheat variety in each depth increment over the duration of the experiment. Error bars represent standard errors (n = 3).



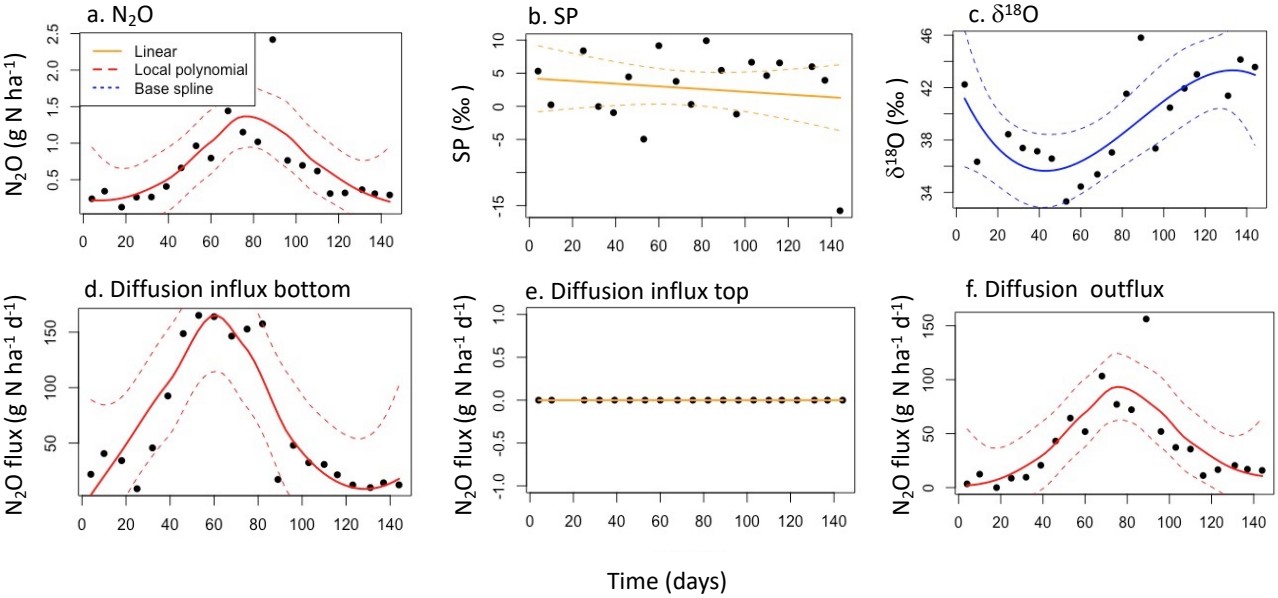

**Figure 2:** Example of fitting smooth curves for temporal patterns in **(a)** $N_2O$ concentrations, **(b)** site preference (SP) values of $N_2O$, **(c)** $\delta^{18}O$
isotope values of $N_2O$, diffusion fluxes **(d)** entering from the bottom and **(e)** top of the depth increment, and **(f)** diffusion fluxes leaving the
depth increment for the 45-75 cm depth increment in one of the columns. The color of the fitted line indicates if the best model was linear
(yellow), a local polynomial (red) or a base spline (blue). Dotted lines indicate 95% confidence intervals.

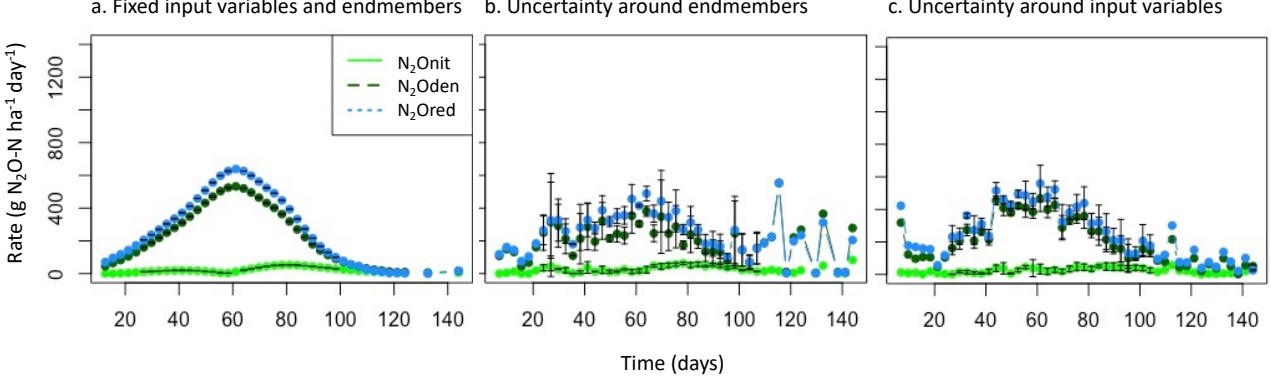

**Figure 3:** Assessment of the effect of uncertainty in isotope endmembers and input variables on modeled nitrification derived $N_2O$
production ($N_2O_{nit}$), denitrification derived $N_2O$ production ($N_2O_{den}$) and $N_2O$ consumption ($N_2O_{red}$), in g $N_2O$-N ha$^{-1}$ day$^{-1}$. Scenarios incude
**(a)** no uncertainty in endmembers and input variables; **(b)** uncertainty in endmembers; and **(c)** uncertainty in input variables. Model output
is shown one depth increment in one lysimeter. Error bars represent the standard deviation around the 25% best parameter estimates over
1000 iterations.





820

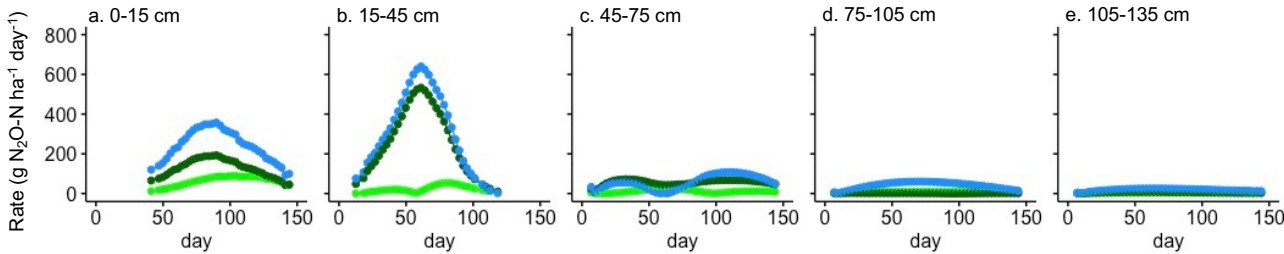

**Figure 4:** Nitrification derived $N_2O$ production rates (light green), denitrification derived $N_2O$ production rates (dark green) and $N_2O$ reduction rates (blue) for each of the soil depth increments in one of the columns.

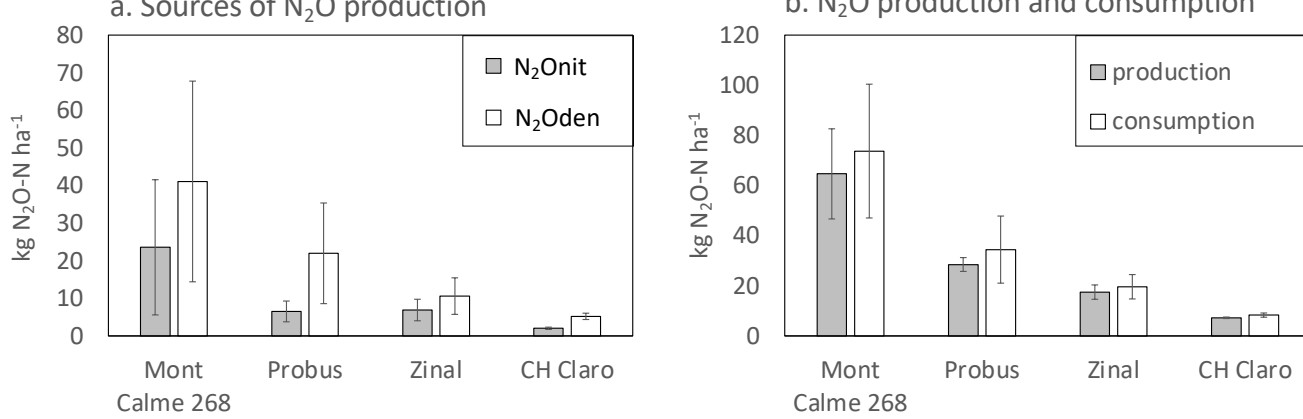

825

**Figure 5: (a)** Cumulative nitrification ($N_2O_{nit}$) and denitrification derived $N_2O$ ($N_2O_{den}$) derived $N_2O$ production across the depth profile and **(b)** total $N_2O$ production and consumption across the depth profile in kg $N_2O$-N $ha^{-1}$ over the course of the experiment.

830