# Peer review of "Process Rate Estimator: A novel model to predict total denitrification using natural abundance stable isotopes of N2O"

_Biogeosciences, 2022_

## Author Comment (AC1)

Eq 7,8 – eta*F reduced/diffused – you make open system assumption in your equations – definitely right for diffusion, but can be doubt for N2O reduction, there might be values which cannot be explained with open system equation (Lewicka-Szczebak et al., 2014, 2017), or you can wrongly attribute this to larger proportion of nitrification. Especially given that you have large reduction it makes big difference if using open or closed system equations. I would check what is the difference and report this. You have only justified using the open system equation with the argument of simplified maths, but I think this generates significant bias. The requirement for using open system equation is the equilibrium of the fluxes in and out (Fry, 2006) and you clearly do not have this between N2O production and reduction.

We acknowledge that carefully considering open versus closed system dynamics is important. If $N_2O$ production and reduction occur simultaneously, both open or closed dynamics could occur. The reviewer refers to the requirement of equilibrium between the fluxes in and out for open system dynamics, and argues this is not the case in our study. We kindly disagree with this assessment. Given the slow but steady change of $N_2O$ and isotope values over time observed in our study, fluxes in and out may well be at near equilibrium in distinct time points. In the study by Lewicka-Szczebak and collegues (2017), instantaneous $N_2O$ emitted from the soil surface was measured during an incubation, where a $N_2O$ pulse was observed following application of fertilizer N. When plotting isotope values of $N_2O$ in function of the residual $N_2O$ fraction, a logarithmic fit was found, which could be described by the Rayleigh equation that describes $N_2O$ fractionation under closed system dynamics. This clearly illustrates that in their study, closed system dynamics prevailed during $N_2O$ reduction. It should be noted, however, that the patterns observed in Lewicka-Szczebak et al. (2017) are only expected to occur if $N_2O$ source processes remain constant, while there is a progressive reduction of $N_2O$ over time. If $N_2O$ production from nitrification increase or decrease over time, very different patterns would have emerged. The figures below show simulated relationships between SP of the instantaneous product and the residual substrate fraction under scenarios where $N_2O$ production is constant over time, vs. where $N_2O$ production from nitrification and denitrification changes over time, for open vs. closed system dynamics. In all scenarios, we assumed that the residual fraction of $N_2O$ decreases progressively over time.

[Figure]

[Figure]

We argue that, while the study by Lewicka-Szczebak et al. (2017) marked an important milestone in furthering how to interpret N$_2$O isotope values, the study does not provide unequivocal evidence that closed system dynamics will prevail in all scenarios where N$_2$O production and reduction occur simultaneously. This leaves the question, when is it appropriate to use open vs. closed system dynamics? In a simulation by Denk et al. 2017, it was shown that the difference between open and closed system dynamics becomes small at small time steps, even when the fraction of residual substrate becomes very small. Given that closed system dynamics would lead to a system of complex non-linear equations that are much harder to solve numerically, we opted to use open system dynamics and a small time step. We will elaborate on this point in the discussion, including recommendations for future research to provide more clarity on this issue.

Denk, T. R. A., et al. (2017). "The nitrogen cycle: A review of isotope effects and isotope modeling approaches." Soil Biology & Biochemistry 105: 121-137. (Fig. 7)

L 555 (Appendix) – "With various terms cancelling out, the equation can be rewritten as:" – why can you cancel out "various things", what are these "various things" and how this cancelling out influences the final result?

The canceling out of terms follows rules in algebra. To make it more clear, we strike through the terms that cancel out below

**Equation A.2:**

$$\frac{\Delta I}{\Delta t} = \big(I_0 C_0 + I_{in,1} k_{in1} \Delta t + I_{in,2} k_{in,2} \Delta t$$
$$- \eta_{out,1}\big(C_0 + k_{in,1}\Delta t + k_{in,2}\Delta t - C_0 - k_{in,1}\Delta t - k_{in,2}\Delta t + k_{out,1}\Delta t\big)$$
$$- \eta_{out,2}\,\big(C_0 + k_{in,1}\Delta t + k_{in,2}\Delta t - C_0 - k_{in,1}\Delta t - k_{in,2}\Delta t + k_{out,2}\Delta t\big) - I_0 C_0$$
$$- I_0 k_{in,1}\Delta t - I_0\, k_{in,2}\Delta t\big)\big)/\big((C_0 + k_{in1}\Delta t + k_{in2}\Delta t)\Delta t\big)$$

**Equation A.2 with strike-through text for terms that cancel out**

$$\frac{\Delta I}{\Delta t} = \big(\cancel{I_0 C_0} + I_{in,1} k_{in1} \Delta t + I_{in,2} k_{in,2} \Delta t$$
$$- \eta_{out,1}\big(\cancel{C_0} + \cancel{k_{in,1}\Delta t} + \cancel{k_{in,2}\Delta t} - \cancel{C_0} - \cancel{k_{in,1}\Delta t} - \cancel{k_{in,2}\Delta t} + k_{out,1}\Delta t\big)$$
$$- \eta_{out,2}\,\big(\cancel{C_0} + \cancel{k_{in,1}\Delta t} + \cancel{k_{in,2}\Delta t} - \cancel{C_0} - \cancel{k_{in,1}\Delta t} - \cancel{k_{in,2}\Delta t} + k_{out,2}\Delta t\big) - \cancel{I_0 C_0}$$
$$- I_0 k_{in,1}\Delta t - I_0\, k_{in,2}\Delta t\big)\big)/\big((C_0 + k_{in1}\Delta t + k_{in2}\Delta t)\Delta t\big)$$

Removing the terms that cancel out result in the following equation:

**Equation A.3:**

$$\frac{\Delta I}{\Delta t} = \frac{\big(I_{in,1} k_{in,1}\Delta t + I_{in,2} k_{in,2}\Delta t - \eta_{out,1} k_{out,1}\Delta t - \eta_{out,2}\, k_{out,2}\Delta t - I_0 k_{in,1}\Delta t - I_0\, k_{in,2}\Delta t\big)}{(C_0 + k_{in,1}\Delta t + k_{in,2}\Delta t)\Delta t}$$

Note that $\Delta t$ appears in every term in the enumerator, and also appears in the denominator. Following algebra, $\Delta t$ cancels out to the following equation:

Equation A.4:

$$\frac{\Delta I}{\Delta t} = \frac{\left(I_{in,1}k_{in,1} + I_{in,2}k_{in,2} - \eta_{out,1}k_{out,1} - \eta_{out,2}\,k_{out,2} - I_0 k_{in,1} - I_0\,k_{in,2}\right)}{\left(C_0 + k_{in,1}\Delta t + k_{in,2}\Delta t\right)}$$

We hope this clarifies the math. There are no assumptions applied here, strictly doing algebra.

L563 (Appendix) – "For $\Delta t$ approaching 0, the equation can be further simplified as:" – why can you assume Dt approaching 0? It is one day, right? When you are using fluxes in your equations you need the time factor to be included in the equations, I guess. What are the units for the fluxes used in your Eq 7 and 8?

The reviewer is referring to the following section of the proof for the equation used in this study:

For $\Delta t$ approaching 0, the equation can be further simplified as:
Equation A.5:

$$\frac{\Delta I}{\Delta t} = \frac{k_{in,1}\left(I_{in,1} - I_0\right) + k_{in,2}\left(I_{in,2} - I_0\right) - \eta_{out,1}k_{out,1} - \eta_{out,2}\,k_{out,2}}{C_0}$$

We argue that $\frac{\Delta I}{\Delta t}$ is considered the infinitesimal change in the isotope value over a short time step. Thus, if $\Delta t$ is very small, $k_{in,1}\Delta t$ and $+ k_{in,2}\Delta t$ would approach 0, and the denominator can be reduced to $C_0$. In our study, we estimate $\frac{\Delta I}{\Delta t}$ as the first derivative to the smooth curve fitted to the isotope values over time. As such, $\frac{\Delta I}{\Delta t}$ does indeed denote the infinitesimal change in the isotope value as the change in time (the time step) approaches 0. Therefore, we argue that our proof of our equation is valid.

---

## Author Comment (AC3)

Equations (8) and (9): These equations are used to reflect both isotopic fractionation and mixing. However, the mixing part is necessarily mass / isotope conserving. I.e., it is only a valid approximation if Ftop, Fbot, N2Onit, N2Oden and N2Ored are small compared to N2Oconc,0. The reason is as follows: Assume that in (7), N2Onit 4 times N2Oconc,0 and SP0 is 0. The shift in SP would be 4 *34.4 per mil = 137.6 per mil. This is outright impossible. Consequently, the formulation as it is doesn't ensure a physically sane mixing process as the highest possible shift would be limited to a shift that results in SP of 34.4 per mil for nitrification-derived N2O and 0 for denitrification. This means, the authors need to find criterions for the time step control or find a formulation that allows simultaneous fractionation and mixing calculations in a mathematically correct way.

**We understand the reviewer's concern, but need to underline that our approach is not a simulation model. Clearly, some gross $N_2O$ production rates in the absence of other processes would lead to unrealistic shifts in the isotope values of $N_2O$ when simply plugging numbers into the equation. The point of solving the set of equation is to identify mathematically plausible values for gross $N_2O$ production and consumption based on observed $N_2O$ concentrations and isotope values. If the equations derived in this study were to be used in a simulation model, we agree that further constraints would be essential. In our approach, however, we believe that constraints would unnecessarily add potential bias and assumptions.**

**We do acknowledge that the assumption used in the last step in the model proof is likely only valid when the incoming $N_2O$ is smaller than the initial N pool with each time step. To address the time-step issue, we will solve the model at different time-steps as part of the model testing and validation. This will be done with the original equation used on our study:**

$$\frac{\Delta I}{\Delta t} = \frac{k_{in,1}(I_{in,1} - I_0) + k_{in,2}(I_{in,2} - I_0) - \eta_{out,1}k_{out,1} - \eta_{out,2}\,k_{out,2}}{C_0}$$

**As well as the equation where the time step is not assumed to approach 0.**

$$\frac{\Delta I}{\Delta t} = \frac{\left(I_{in,1}k_{in,1} + I_{in,2}k_{in,2} - \eta_{out,1}k_{out,1} - \eta_{out,2}\,k_{out,2} - I_0 k_{in,1} - I_0\,k_{in,2}\right)}{\left(C_0 + k_{in,1}\Delta t + k_{in,2}\Delta t\right)}$$

**Final model results for the whole data set will be shown for the model and time step that rises as the best performing in our sensitivity analysis for the time-step optimization.**